# Comprehensive SHAP Values and Single-Cell Sequencing Technology Reveal Key Cell Clusters in Bovine Skeletal Muscle

**DOI:** 10.3390/ijms26052054

**Published:** 2025-02-26

**Authors:** Yaqiang Guo, Fengying Ma, Peipei Li, Lili Guo, Zaixia Liu, Chenxi Huo, Caixia Shi, Lin Zhu, Mingjuan Gu, Risu Na, Wenguang Zhang

**Affiliations:** 1College of Animal Science, Inner Mongolia Agricultural University, Hohhot 010010, China; gggyaqiang@163.com (Y.G.); fengyingma1997@163.com (F.M.); 13474912747@163.com (L.G.); 15660097986@163.com (C.H.); shicx98@163.com (C.S.); zhulinynacxhs@163.com (L.Z.); gmj0119@yeah.net (M.G.); 2Inner Mongolia Engineering Research Center of Genomic Big Data for Agriculture, Hohhot 010018, China; 3College of Life Sciences, Inner Mongolia University, Hohhot 010020, China; 111992071@imu.edu.cn

**Keywords:** single-cell RNA-sequencing, cattle, skeletal muscles, machine learning, SHAP

## Abstract

The skeletal muscle of cattle is the main component of their muscular system, responsible for supporting and movement functions. However, there are still many unknown areas regarding the ranking of the importance of different types of cell populations within it. This study conducted in-depth research and made a series of significant findings. First, we trained 15 *bovine* skeletal muscle models and selected the best-performing model as the initial model. Based on the SHAP (Shapley Additive exPlanations) analysis of this initial model, we obtained the SHAP values of 476 important genes. Using the contributions of these 476 genes, we reconstructed a 476-gene SHAP value matrix, and relying solely on the interactions among these 476 genes, successfully mapped the single-cell atlas of *bovine* skeletal muscle. After retraining the model and further interpretation, we found that Myofiber cells are the most representative cell type in *bovine* skeletal muscle, followed by neutrophils. By determining the key genes of each cell type through SHAP values, we conducted analyses on the correlations among key genes and between cells for Myofiber cells, revealing the critical role these genes play in muscle growth and development. Further, by using protein language models, we performed cross-species comparisons between cattle and pigs, deepening our understanding of Myofiber cells as key cells in skeletal muscle, and exploring the common regulatory mechanisms of muscle development across species.

## 1. Introduction

*Bovine* skeletal muscle occupies a central position within the *bovine* body, supporting a variety of movements and life demands through its intricate structure and functional mechanisms [1,2,3]. The formation of muscle commences from the embryonic stage, progressing through the development of primary and secondary muscle fibers, before gradually decelerating in the late stages of gestation [4,5]. The proliferation and hypertrophy of skeletal muscle fibers significantly augment muscle mass, and the developmental period of livestock skeletal muscle plays a decisive role in its future muscular development [6,7]. Muscle formation is a complex process involving the concerted action of multiple cell types and gene networks [8,9,10]. Additionally, skeletal muscle exerts a significant defensive role in bodily health, with its contraction activities upregulating antioxidant defenses, safeguarding the proliferative potential of T-cells from inflammation, and replenishing effector cells to counter T-cell exhaustion [11,12]. A close association exists between low skeletal muscle mass, sarcopenia, and sarcopenic obesity [13], underscoring the urgent need to understand their regulatory mechanisms.

However, many mysteries remain regarding the ranking of the importance of skeletal muscle-dominant cells, the critical genes of different cell types, and the interactions among these genes. In recent years, with the continuous emergence of large language models and the widespread application of machine learning, particularly its exceptional performance in data classification and regression problems, machine learning has become a focal point of research [14,15]. Notably, random forests and deep learning, as significant branches within the field of machine learning, each exhibit unique advantages [16]. Random forests excel in handling small sample data, efficiently processing datasets with numerous predictor variables, and boasting superior classification accuracy [17,18,19]. SHAP, a model interpretation method based on cooperative game theory, quantifies the contribution of each feature to the model’s output, elucidating how model predictions are determined by feature values [20]. Deep learning, modeled after the structure of the human brain’s neural networks, demonstrates notable advantages in handling complex data, automatic feature extraction, adapting to large-scale data, and continuous learning capabilities [21,22,23]. The confluence of these three approaches presents an opportunity to address this issue.

Advances in single-cell sequencing technologies have afforded us unprecedented opportunities, allowing for a high-resolution dissection and functional state evaluation of cellular heterogeneity by analyzing genetic information within individual cells. This has shed light on the diversity of cells and their roles in organismal development and disease [24]. In this study, we leveraged publicly available single-cell data from *bovine* and *porcine* skeletal muscle to construct a skeletal muscle single-cell model based on the Random Forest (RF) algorithm. Using this model, we successfully mapped the single-cell atlas of *bovine* skeletal muscle, determining the importance rankings of various skeletal muscle cell types and the distribution of key genes. Specifically, taking Myofiber cells as an example, we conducted an in-depth analysis by comparing them with the next-ranked neutrophils and FAP cells, which are closely related to fat development, revealing the crucial roles of key genes, such as *ACTA1*, *CKM*, and *MYOZ1,* in the physiological functions of skeletal muscle.

To further validate and deepen these findings, we integrated the SATURN [25] protein language model, which is based on deep learning, to conduct a cross-species comparative analysis of skeletal muscle in both cattle and pigs. Through an in-depth study of Macrogenes data, we found that the results from both analytical approaches exhibited high consistency, further strengthening the validity of our research conclusions.

## 2. Results

### 2.1. Construction of the Primitive Atlas for Bovine Skeletal Musculature

In this study, the *bovine* skeletal muscle single-cell dataset used comprises fourteen types of cells, with a total of 32,708 cells, among which FAP cells account for more than one-third of the total (Figure 1A), and there are only 44 neutrophils. Additionally, the average expression levels for the top 10% to bottom 10% of each level were calculated for each cell type (Figure 1B). Remarkably, the top 10% of genes in each cell type account for nearly 50% or more of the total expression. However, this cannot definitively designate these genes as key genes for the fourteen skeletal muscle cell types, as this represents only a macro expression pattern and fails to dissect the complex relationships between genes. For the original cell expression matrix, a single-cell map of *bovine* skeletal muscle cells has been mapped based on the initial number of cells and genes (Figure 1C).

### 2.2. Multi-Model Optimization and the Identification of Key Genes in Skeletal Muscle

In the process of data fitting, we use the number of neutrophils as a benchmark, performing under-sampling on the other 13 cell types that have counts greater than that of neutrophils. In the subsequent 12 model training sessions, for cells with counts larger than the target cell count, we perform under-sampling, while for those with counts smaller than the target, we perform oversampling. In the 15th training session, all cells with counts lower than that of the FAP cells are subject to oversampling (Figure 2A). Therefore, we trained 15 single-cell models (A model trained using raw data as input, and 14 models trained after fitting the data based on the quantities of 14 types of cells) of *bovine* skeletal muscle. One of them was a model trained without fitting data, and the remaining 14 were data-fitting models. These 14 models were based on the numbers of 14 different types of cells, respectively. For each model, under-sampling, oversampling, or a combination of oversampling and under-sampling were performed on other types of cells so that the numbers of cells tended to be consistent.

By comparing the performance of the 15 models, we found that the model trained on the original data achieved a Best Cross-Validation Score (BVS) of 0.971 and an Accuracy with Best Parameters (ABPs) of 0.970 in the test set, indicating a good learning effect of the data (Figure 2B). Therefore, we selected the model trained with B cell counts as the benchmark, which had a BVS of 0.969 and an ABP of 0.968 in the test set.

During model training, when the input consists of raw, unfitted data, the model is called the raw data model. After fitting the data with the number of B cells, the trained model is referred to as the initial model. When comparing the original data model with the initial model, in most cell types, the F1 score and recall of the fitted data were superior to those of the unfitted original data. We ultimately recognized the model trained with B cell counts as the initial model and saved the cell atlases of both the training and test sets under the condition of the highest accuracy (Figure 2C,D).

Subsequently, we utilized SHAP values to interpret the initial model, obtaining the top 20 most important genes in *bovine* skeletal muscle (Figure 2E). By analyzing the correlations between these genes, we found that only eight pairs of genes had a correlation greater than 0.8, with most genes showing negative correlations, but with low negative correlation values, mostly not exceeding −0.2 (Figure 2F).

### 2.3. Reconstruction of Single-Cell Atlases and Model Optimization

We extracted the top 500 most important genes from 14 cell types and found that there were only 1161 uniquely expressed genes (Figure 3A), 180 co-expressed genes, and 296 specifically expressed genes. On the selected test set, we trained models using 180, 296, 1161, and the combination of the first two with 476 genes (Figure 3B). Since the four sub-training models were based on the initial model, and the 476 genes included both co-expressed and specifically expressed genes across cells, the parameters of the *bovine* skeletal muscle model constructed with these genes were close to those of the initial model. Therefore, we selected this as the gene model of choice, which we refer to as the 476-gene model. We generated skeletal muscle cell atlases based on these four sets of gene numbers (Figure 3C–F), and the UMAP results were similar to the parameter evaluations of the four models. It can be clearly observed from Figure 3C–F that, in terms of the degree of discrimination of each cell type after dimensionality reduction, for the vast majority of cells in Figure 3E,F, cells of the same type are clustered together. Although the clustering levels of several cell types in Figure 3F are not very obvious, it is evident that the 14 cell types are well-distinguished by the expression levels of the 476-gene and 1161-gene. In Figure 3C,D and in Figure 3C, four cell types in the middle area are too closely connected. In Figure 3D, only four cell types are clearly clustered, and it is difficult to distinguish the remaining ten cell types. Ultimately, we chose the best-performing 476 genes for subsequent analysis.

Through cell composition analysis, we ranked pathways based on gene signal intensity (Figure 3G) and found that the top four cell components were closely related to the development and differentiation of muscle cells. However, the number of genes enriched in these components was relatively smaller compared to the latter two components.

### 2.4. Construction of Single-Cell Atlas and Identification of Key Genes Based on SHAP Value Optimization

After obtaining the SHAP values of the initial model, we utilized the additivity of SHAP values to reconstruct the SHAP values of the selected 476 genes with all the correctly classified cells in the test set, resulting in a SHAP value matrix for the 476 genes (Appendix A). Through matrix reconstruction and retraining the model, the model was evaluated (Figure 4A). The accuracy of the model was close to one, meeting our expectations. We inferred that this method successfully identified more than 99% of the important information in the cells.

We summed the SHAP values greater than 0 for the 476 genes in each cell type, and after removing low-quality information, we obtained the importance ranking of 14 cell types (Figure 4B). Among them, Myofiber cells were the most representative cells of skeletal muscle, followed by neutrophils. Based on the SHAP value matrix of the 476 genes, we plotted the single-cell atlas of skeletal muscle (Figure 4C), where all 14 cell types were well distinguished.

Through the contribution of genes to each cell type, we identified the top 20 key genes for each cell type(Appendix A). Taking Myofiber cells as an example, we conducted a preliminary analysis of the top 20 genes (Figure 4D) and found that the most representative gene was *ACTA1*. These genes had the highest expression correlation in Myofiber cells, followed by neutrophils, and their expression correlation significantly decreased in the other 12 cell types (Figure 4E). Next, we performed biological process and tissue expression analysis on the top 20 genes ranked by SHAP values in Myofiber cells. In biological processes, 15 genes were enriched in seven pathways, with five pathways closely related to muscle growth and development, and another 5 genes enriched in the development of skeletal muscle tissues and organs (Figure 4F). In terms of gene expression in tissues, 16 genes were enriched in 10 tissues, including five heart tissues and four muscle tissues, such as skeletal muscle and longissimus dorsi (Figure 4G).

### 2.5. Comparative Analysis of Gene Expression and Functional Characteristics in Myofibers Relative to Other Cellular Types

To verify the scientific significance of key pre-two cell types in *bovine* skeletal muscle, myofibers, and neutrophils, we compared these two types of cells with FAP cells to enhance the persuasive power of the research results. The reason for choosing FAP cells lies in their characteristics closely related to adipogenesis, forming a sharp contrast to the myogenesis effect of myofiber cells. After integrating the top 20 genes ranking of these three cell types, we obtained 47 uniquely expressed genes and the correlation analysis results with cell types (Figure 5A). Further analysis of the biological processes of the top 10 genes of these three cell types showed that 18 genes were enriched in eight pathways, of which four pathways were closely related to muscle tissue, and the majority were related to immune processes (Figure 5B). In the analysis of the top 20 genes ranked by SHAP values for 14 cell types, we did not find any gene co-expressed (Figure 6C). While FAP, neutrophils, and myofibers specifically expressed 1, 3, and 11 genes, respectively. Correlation analysis of these 15 genes showed that 27 gene pairs had correlations greater than 0.8, with MYOZ1 and MYL1 being particularly prominent, forming 8 and 9 highly correlated gene pairs with other genes, respectively (Figure 5D). In further gene–cell correlation analysis, we found more genes highly correlated with myofibers and neutrophils (Figure 5E,F).

Finally, we conducted an enrichment analysis of biological processes and tissue expression for these 15 genes. The results showed that nine genes were enriched in four biological processes closely related to muscle growth and development (Figure 5G); at the same time, these genes showed high expression in three main tissues—the heart, trapezius, and skeletal muscle (Figure 5H).

### 2.6. Cross-Species Single-Cell Analysis of Bovine and Porcine Skeletal Muscle

The skeletal muscle of pigs encompasses eight types of cells, among which the number of Myofiber cells is the largest, reaching 30,328 (Appendix A). We followed the species protein sequence training script on its official website and input the protein sequences of cattle and pigs to train the corresponding protein models for the two species. Then, we used the official training script of SATURN to train the model with the generated protein embeddings from the training, the gene expression matrices of the two species, and the initial intraspecies cell annotations. The parameters of the original script in both training sessions did not change. We set the number of highly variable genes to 8000 and the number of macrogenes to 2000.

Skeletal muscle cells of cows and pigs exhibit certain genetic conservatism in terms of species and cell types, showing an inverted Y-shaped similarity (Figure 6A). However, in terms of specific cell type expression, Muc and FAP cells exhibit significant differences between the two species, while the expression of pig myofiber cells is significantly higher than that of cow myofiber cells, almost obscuring the latter’s expression characteristics (Figure 6B). It can be seen that the Myofiber cells of cattle did not show data related to the skeletal muscles of pigs. Among the cells of the input species, the number of pig Myofiber cells is 30,328, while there are only more than 500 in cattle. Our speculation is that the excessive number of such cells in pigs may have masked the expression of such cells in cattle. Among the significantly correlated metagenes of myofiber cells, we identified metagene 1537 (Figure 6C). Further enrichment analysis of the top 20 genes within the metagenes of the two species revealed that 17 genes were enriched in four major cellular components, with 6 genes enriched in supramolecular fibers, and 9 genes enriched in muscle fibers (Figure 6D).

After a thorough analysis of the metagenes of the other 12 cell types and one unknown cell type, we first counted the number of genes with weights greater than one under each cell type’s metagene. To ensure the accuracy of the analysis, we established strict screening criteria: if the number of genes greater than one was less than three, only the top three or four genes were counted. These genes were identified as the main effect genes of each cell type (Appendix A). Next, we studied the correlation between the expression level of each cell type and its metagene (Figure 6E). The results showed that neutrophils had a significantly higher expression level with their metagene 201 than other cells and had the most metagenes, even significantly higher than the metagene levels of CD8 T cells and natural killer cells within their own cell types. This finding further confirms the reason why neutrophils rank high among the cell populations of cow skeletal muscle. We speculate that neutrophils may have a high degree of synergy among immune cells and occupy a dominant position.

To gain a deeper understanding of the metagenomic characteristics of each cell type, we statistically analyzed the top 20 genes and their weights under each cell type’s metagene (Appendix A). Summing the weights of the top 20 genes for each type, we found that neutrophils had the highest total weight (Figure 6F).

## 3. Discussion

In this study, we comprehensively explored *bovine* skeletal muscle single-cell transcriptomic data through the integration of explainable machine learning models with open-source protein large models, and successfully conducted a cross-species comparison with *porcine* skeletal muscle single-cell transcriptomes. It is crucial to emphasize that the current mainstream single-cell sequencing technologies, represented by 10× single-cell sequencing, have significant limitations in information extraction. These technologies are only capable of capturing approximately 30–40% of the cellular information, which means that at least 60% of the essential data are lost [26,27]. This substantial data loss not only severely degrades the data quality but also makes the information obtained from less-expressed cell types even sparser [28]. In the context of our current research, such a high proportion of data loss implies that a large amount of potentially valuable biological information may be overlooked. This could have a profound impact on our conclusions, as the missing data might contain key regulatory factors, rare cell populations, or subtle expression patterns that are essential for a comprehensive understanding of the biological processes in *bovine* and *porcine* skeletal muscles. Therefore, while our findings are significant, they should be interpreted with caution, taking into account the inherent limitations of the single-cell sequencing technology employed.

To address this challenge, we employed the SMOTE technique, which expands the representation of underrepresented samples without losing critical information, while simultaneously mitigating the biological disparities between excessive samples and minority classes through under-sampling [29,30]. By comparing the results of models trained on the fitted data with those trained on the original data, we confirmed the suitability of the data fitting approach. Although the fitted data offer numerous advantages, their drawbacks are also prominent, particularly the tendency towards overfitting, which can lead to model misclassification. By keeping the accuracy of the fitted data within a range approximating that of the original data, this study successfully averted this issue.

Previous studies have largely relied on prior research experience to analyze genes or cell types of interest, yielding results albeit often devoid of logical and scientific rigor [31,32,33]. According to the Pareto Principle, about 80% of the effects come from 20% of the causes. In genetic research, this could mean that only a small portion of a large number of genes produce the primary impact [34,35]. Building upon the methodologies of these earlier studies, we scientifically identified 476 significant genes from tens of thousands of cells and gene expressions. Utilizing the SHAP values, we reconstructed the SHAP value matrices for these 476 genes, which not only quantified the relationships between cells and genes but also elucidated the interactions between genes. By leveraging the SHAP matrix of these 476 genes, we strengthened the associations of the selected genes, effectively excluding weakly, non-, and negatively correlated factors, thereby constructing a comprehensive single-cell atlas of *bovine* skeletal muscle. Subsequently, through SHAP interpretation, we obtained rankings of cell importance and key genes based on the SHAP values.

We have identified Myofibers as representative cells of skeletal muscle, with neutrophils following closely behind. Myofibers are composed of highly oriented muscle fibers, and their contractile properties dictate the morphology and function of skeletal muscle, such as promoting the regeneration of functional muscle, maintaining muscle strength, and force [36,37,38]. Neutrophils play a crucial role in the repair process of skeletal muscle injury and function as guardians in immune surveillance, particularly in muscles that have lost their neural innervation [39,40]. Studies indicate that neutrophils can enhance skeletal muscle function during physical activity, and their aging in elderly mice can induce muscle inflammation [41]. These findings underscore the critical importance of Myofibers and neutrophils to the function and homeostasis of skeletal muscle.

Regarding the pivotal genes of Myofibers, we have discovered that *ACTA1*, the skeletal muscle alpha-actin, ranks first [42]. This protein is central to the thin filaments of the sarcomere in skeletal muscle and is crucial for muscle contraction in conjunction with myosin [43,44]. Through bio-functional and tissue expression analysis, we found that the majority of the key genes are significantly enriched in muscle growth and development, as well as in muscle and cardiac tissues. This indicates the soundness of our research methodology and the scientific validity of our findings.

By comparing the key genes across different cell types, we observed that Myofibers exhibit a greater number of specifically expressed genes compared to FAPs and neutrophils, with the correlation coefficients among these genes mostly exceeding 0.8. This indirectly suggests the unique functional roles of Myofibers. In subsequent interspecies comparisons, although the macro-gene weight of neutrophils significantly surpasses that of FAPs and Myofibers, our correlation analysis reveals that neutrophils not only possess a high number of Macrogenes but also exhibit elevated weights across different cell types, suggesting robust communication with other cells and possible dominance in their functional aspects. Moreover, an analysis of the highly weighted genes under macro-genes, using Myofiber cells as an example, shows that a considerable number of genes are again enriched in muscle growth, developmental biology, and muscle tissue. The striking consistency between these two entirely different research approaches underscores the scientific rigor and methodological soundness of our study.

Although this study has achieved certain results, there are limitations that cannot be ignored. In terms of sampling technology, there may be potential biases. The sample selection process may not have fully covered the diversity of the target population, resulting in insufficient representativeness. This may limit the generalizability of the research results and prevent them from accurately reflecting the overall situation. The machine-learning model has the risk of overfitting. The model performs well on the training set and test set data, but it is very likely to have poor predictive ability for new data, which may weaken the reliability and practicality of the research conclusions. The single-cell sequencing technology itself also has its limitations. For example, differences in cell capture efficiency may lead to some cell types not being fully detected. Uneven sequencing depth may affect the accuracy of gene expression quantification. Moreover, the high cost of this technology and the complexity of data analysis also restrict the breadth and depth of the research to a certain extent. Subsequent research needs to focus on improving these issues to enhance the quality of the research and the credibility of the results.

## 4. Materials and Methods

### 4.1. Data Sources

The *bovine* skeletal muscle single-cell transcriptome data used in this study were sourced from the GEO public database [45], with accession number GSE205347. The research data were directly downloaded from the database in the cell and gene matrix data format. Specifically, the *bovine* data included three breeds: Kobe cattle, Brahman cattle, and a crossbreed between Kobe and Brahman cattle. *Porcine* skeletal muscle single-cell data were sourced from URL (https://ngdc.cncb.ac.cn/bioproject/browse/PRJCA017014, accessed on 9 November 2024), derived from the longissimus dorsi muscle of a boar.

### 4.2. Training of Bovine Skeletal Muscle Models

In this experiment, we employed a suite of Python [46] toolkits to meticulously categorize the *bovine* skeletal muscle data. During the training process of the *bovine* skeletal muscle model, we set the features as genes, and the target was 14 types of cells. Properly speaking, the model we trained was a classification model, that is, to classify cells based on the gene expression status. Initially, we utilized pandas (v2.1.4) to process and stabilize the data, ensuring balanced cell counts. This process was achieved through under-sampling and oversampling techniques from the imbalanced-learn library (v0.12.3). Subsequently, the raw data were bifurcated into training and test sets in a 7:3 ratio, with a random seed set at 42 to ensure experimental reproducibility. During the model training, we used several major Python libraries: scikit-learn (v1.5.1), matplotlib (v3.8.0), and seaborn (v0.12.2).

The items for setting the parameters and hyperparameters of the model are as follows: First, to ensure the reproducibility of the experimental results, we choose the random seed as 42 during the process of dividing the training set and the test set and the training process of the model. When using RandomizedSearchCV to perform hyperparameter tuning on the random forest model rf_model, the following hyperparameter distributions are defined. n_estimators randomly selects an integer from 100 to 500, which represents the number of decision trees in the random forest; max_features can take the values ‘auto’, ‘sqrt’, or ‘log2’, which is used to determine the calculation method of the maximum number of features considered when building each tree; max_depth randomly selects an integer from 10 to 50, which controls the maximum depth of the decision tree; min_samples_split randomly selects an integer from 2 to 10, representing the minimum number of samples required to split an internal node; min_samples_leaf randomly selects an integer from 1 to 4, which is the minimum number of samples required for a leaf node; bootstrap is a boolean value that can take the values True or False, used to decide whether to use bootstrapping when building the trees. Based on these hyperparameter distributions, a RandomizedSearchCV object is created. The number of iterations for the random search is set to 20 times, and 5-fold cross-validation is adopted.

### 4.3. Analysis of SHAP Explainability in Skeletal Muscle Models

To enhance the transparency and interpretability of the model, we utilized the SHAP library (v0.44.0) to perform SHAP value analysis on a pre-trained random forest model, calculating SHAP values for each sample based on test set data. During this process, we employed the numpy (v1.21.2) and pandas (v2.1.4) libraries to handle and store these SHAP values. The specific methodology involved initially creating an empty DataFrame list, dfs, and then iterating through each category, filtering correctly classified samples from the test set, and calculating the corresponding SHAP values. Subsequently, SHAP values for each category were converted into DataFrame format, with a column added to denote the category. Finally, all category DataFrames were merged into a comprehensive DataFrame.

### 4.4. Model Training for 476 Co-Expressed and Specifically Expressed Genes

In further research, we based our analysis on the SHAP value matrix of the previously trained model, selecting the SHAP value matrix for 476 specific genes and analyzing them across all correctly classified cell types. During both model training and interpretability analysis, we maintained the same techniques and parameter settings to ensure the continuity and consistency of the study.

### 4.5. Cross-Species Comparison of Skeletal Muscle in Cattle and Pigs

In the realm of cross-species comparisons, we have specifically introduced the open-source protein language model—SATURN, which adeptly addresses the challenge of homologous gene alignment across different species through deep learning techniques. SATURN, having been trained on protein sequence data from multiple species, is capable of quantifying the weight of genes from various species within specific cell types from a Macrogenes perspective, thereby enabling precise cross-species alignment of single-cell transcriptomes. Notably, SATURN incorporates optimization procedures, including data normalization and batch effect removal, allowing for the direct use of raw single-cell transcriptome matrices as input.

Given gene expression and protein embeddings, SATURN learns an interpretable feature space shared between multiple species. This space is a macrogene space and it represents a joint space composed of genes inferred to be functionally related based on the similarity of their protein embeddings. The importance of a gene to a macrogene is defined by neural network weight—the stronger the importance, the higher the value of the weight that connects the gene to the macrogene.

For detailed instructions on using SATURN, the required libraries, and their versions, please visit its GitHub page URL (https://github.com/snap-stanford/SATURN, accessed on 15 November 2024). The protein sequence data for cattle and pigs were sourced from the Ensembl [47] database URL (https://www.ensembl.org/, accessed on 20 November 2024), with the sequence data for cattle available at URL (https://ftp.ensembl.org/pub/release-113/fasta/bos_taurus/pep/, accessed on 15 September 2024), and the sequence data for pigs at URL (https://ftp.ensembl.org/pub/release-113/fasta/sus_scrofa/pep/, accessed on 15 September 2024).

### 4.6. Data Visualization

In this study, we comprehensively elucidated and analyzed the rich nuances of skeletal muscle single-cell transcriptomic data through multi-layered data visualization techniques. Initially, we extensively utilized the scanpy library in Python for single-cell atlas and specific gene expression patterns. Scanpy offers efficient tools for cell clustering, trajectory inference, and visualization, enabling us to extract structured and biologically relevant features from high-dimensional transcriptomic data.

For molecular network and interaction analysis, we employed the STRING [48] online analysis platform URL (https://www.string-db.org/, accessed on 25 November 2024). During the usage process, we inputted the selected gene names and the corresponding species. We conducted pathway enrichment analysis of the genes based on the gene signal intensity of the genes in the species and their expression in tissues.

Before visualizing the correlation relationships between the selected genes, we calculated the Pearson correlation coefficients between the genes using the corr() method in pandas and obtained a correlation matrix named gene_correlation_matrix. Each element in the matrix represents the linear correlation between the corresponding two genes. In this study, a gene-to-gene correlation greater than 0.8 is defined as a high correlation.

For the analysis of the co-expression and specific expression of genes across cell types, we opted for the upset package in R (v4.3.2) [49] language.

Finally, we extensively employed Excel [50] tools for visualization and report generation.

## 5. Conclusions

In summary, our research unveils the paramount significance of *bovine* skeletal muscle cells and the critical genes of various cell types, meticulously mapping their single-cell landscapes through hundreds of pivotal genes, thus offering novel avenues and methodologies for future studies on *bovine* skeletal muscle. To elucidate the functions of these key genes and to rank the importance of cells, further intensive experimental investigations are imperative to explore their biological roles and developmental potentials.

## Figures and Tables

**Figure 1 ijms-26-02054-f001:**
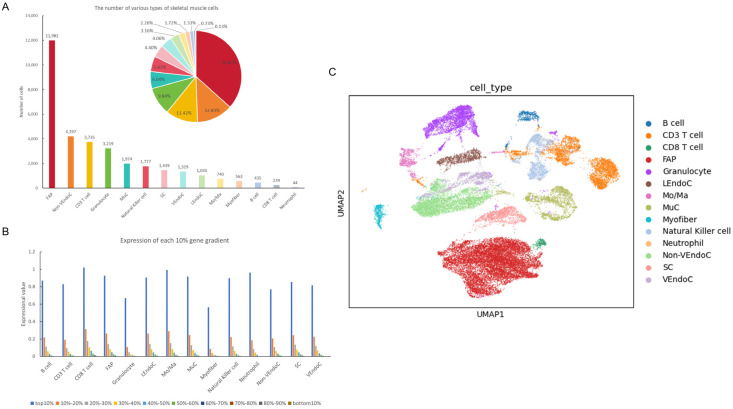
Initial data analysis. (**A**) Statistical evaluation of the number of 14 cell types in *bovine* skeletal muscle. The abbreviated cells in the figure are as follows: FAP (fibro/adipogenic progenitor); VEndoC (venular endothelial cell); MuC (mural cell); MC (myogenic cell); LEndoc (lymphatic endothelial cell); Mo/Ma (monocyte/macrophage). (**B**) Quantitative analysis of gene expression levels, categorizing and averaging the expression in 10% incremental gradients. (**C**) Original matrix-based expression atlas of *bovine* skeletal muscle data.

**Figure 2 ijms-26-02054-f002:**
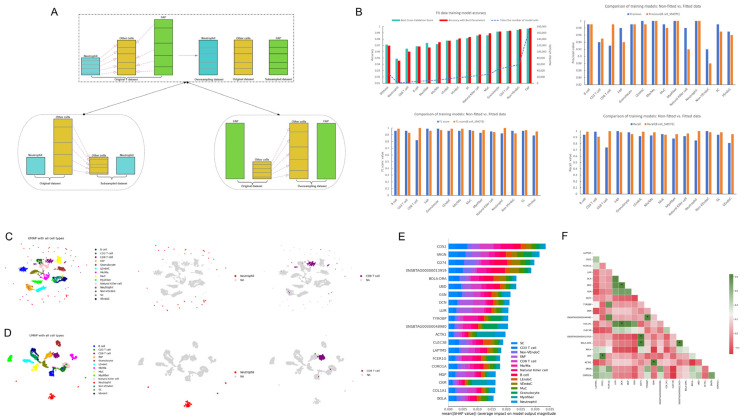
Expression data fitting and model training. (**A**) Schematic depicting the process of data fitting, beginning with the under-sampling of the cell type with the least quantities and progressing to the oversampling of the cell type with the greatest quantities. (**B**) The top left illustrates the accuracy assessment of 15 model trainings, alongside the total number of cells input into the model, while the top right, bottom left, and bottom right, respectively, contrast the precision, F1 score, and recall of the unfitted original data against those of the initial best model. (**C**) Under the condition of the best performance of the initial best model, the training set comprises single-cell maps of skeletal muscle, neutrophils, and CD8 T cells. (**D**) Similarly, under the condition of the best performance of the initial best model, the test set includes single-cell maps of skeletal muscle, neutrophils, and CD8 T cells. (**E**) Following the SHAP interpretation of the initial best model, the top 20 genes by SHAP value are identified. (**F**) Correlation analysis of the top 20 genes, with * indicating gene-to-gene correlations exceeding 0.8.

**Figure 3 ijms-26-02054-f003:**
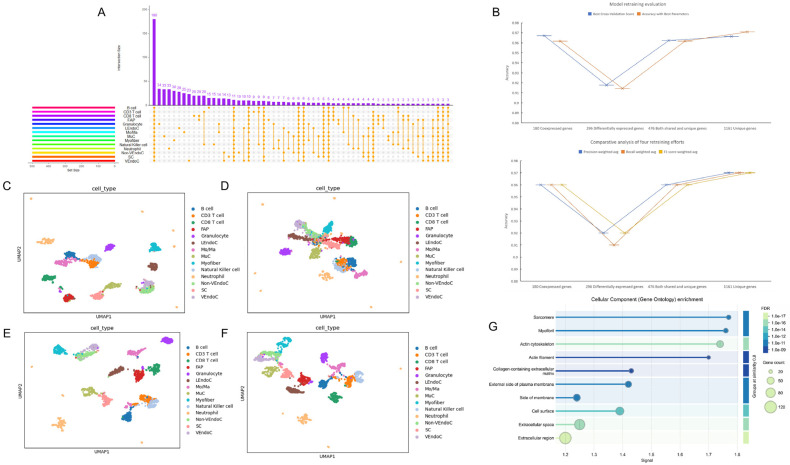
The scientific nature of the training model for 476 co-expressed and specifically expressed genes in 14 types of cells. (**A**) Post SHAP interpretation of the initial optimal model, analysis of the top 500 SHAP-valued genes for co-expression, and specific expression across each cell type. (**B**) Assessment of accuracy following the training of models using matrices composed of 180 co-expressed genes, 296 cell-specific expressed genes, 476 genes from both categories, and 1161 unique genes on the optimal test set of the initial best model. (**C**) Single-cell map of skeletal muscle for 180 genes. (**D**) Single-cell map of skeletal muscle for 296 genes. (**E**) Single-cell map of skeletal muscle for 476 genes. (**F**) Single-cell map of skeletal muscle for 1161 genes. (**G**) Analysis of cellular composition for 476 genes.

**Figure 4 ijms-26-02054-f004:**
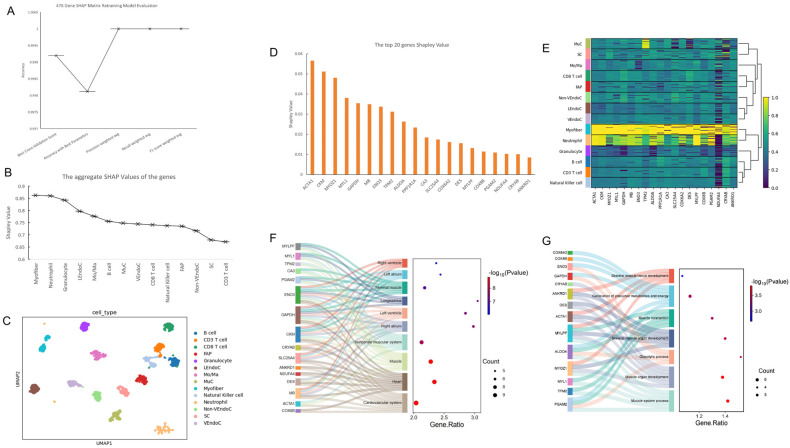
Final mapping of the single-cell profiles of *bovine* skeletal muscle and the identification of key genes and cell types. (**A**) Evaluation of the accuracy of the training model after reconstructing the SHAP matrix with 476 genes. (**B**) Ranking of the importance of 14 cell types in *bovine* skeletal muscle. (**C**) Final determined map of single-cell profiles in *bovine* skeletal muscle. (**D**) Ranking of the top 20 key genes for the most critical cell type, Myofiber, in skeletal muscle. (**E**) Correlation analysis between the top 20 key genes of Myofiber and the 14 cell types. (**F**) Expression levels of the top 20 key genes across the 14 cell types. (**G**) Biological process analysis of the top 20 key genes of Myofiber, with gene names on the left and pathway names in the middle.

**Figure 5 ijms-26-02054-f005:**
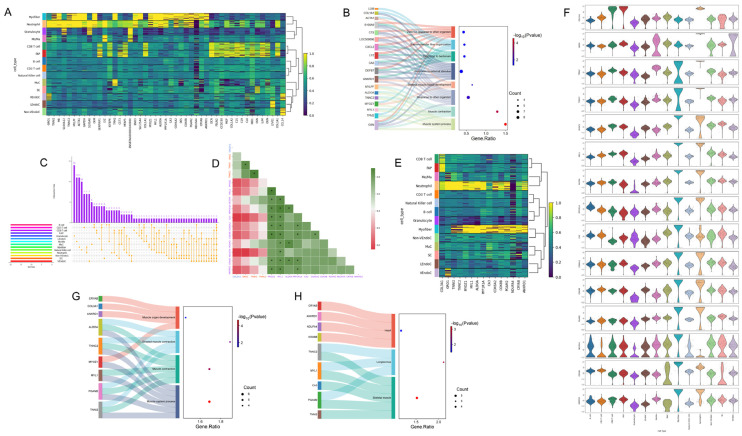
We delve into the intracellular and intercellular information extraction of selected cell types. (**A**) This section delineates the correlation analysis between genes and intercellular interactions among the top 20 key genes in neutrophils, myofibers, and FAPs. (**B**) It further explores the biological processes associated with the top 10 key genes across three categories of cells. (**C**) An analysis of co-expression and specific expression patterns is conducted for the top 20 key genes among 14 categories of cells. (**D**) The correlation between 15 specific expression genes across three cell types is scrutinized, with * indicating correlations greater than 0.8. (**E**) A correlation analysis is performed between these 15 specific expression genes and cellular interactions. (**F**) Violin plots display the expression levels of these 15 specific expression genes across each cell type. (**G**) The biological processes related to these 15 specific expression genes are examined in detail. (**H**) Lastly, the tissue expression analysis of these 15 specific expression genes is presented.

**Figure 6 ijms-26-02054-f006:**
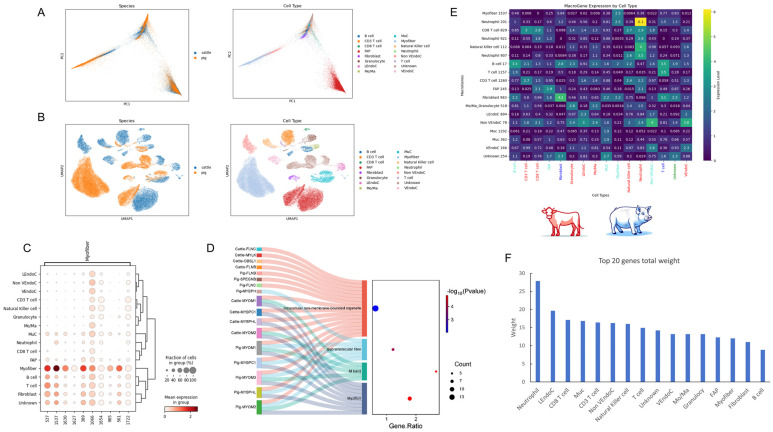
A comparative analysis of skeletal muscle across species and within Macrogene contexts is presented. (**A**) The left panel delineates the principal component analysis spectra of skeletal muscle across species, while the right panel illustrates the PCA spectra for cell types. (**B**) The left panel exhibits the UMAP profiles of skeletal muscle cells across species, with the right panel detailing the UMAP profiles for cell types. (**C**) The heatmap depicts the expression profiles of Macrogene markers for myofibers. (**D**) The cell component analysis of the top 20 genes with the highest weight under the Macrogene context of myofibers is presented. (**E**) A correlation analysis between Macrogene markers and cell types is conducted, where the horizontal axis categorizes cell types by color: red for *bovine*-specific, blue for *porcine*-specific, turquoise for common cell types across both species, and green for unidentified cell types. (**F**) The cumulative weight statistics of the top 20 genes under each cell type’s Macrogene context are summarized.

## Data Availability

The data used in this study all come from publicly available datasets, which are cited in the Section 4. This study did not generate original code.

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
