# Peer review of "Comprehensive SHAP Values and Single-Cell Sequencing Technology Reveal Key Cell Clusters in Bovine Skeletal Muscle"

_ijms, 2025, doi:10.3390/ijms26052054_

Round 1
Reviewer 1 Report
Comments and Suggestions for Authors
Guo and colleagues leveraged the publicly available bovine single-cell RNA sequencing data to train random forest models to identify important genes that differentiate cell types using SHAP values. The authors mapped the cell types with the subset of important genes and identified the representative cell type in skeletal muscle.
This study offers a generalizable framework for studying gene importance and provides insights into understanding skeletal muscle cells. That said, I have a few comments.
Major comments:
-
The resolution of the figures is too low to read the text.
-
The training scheme was not clearly described:
-
The training features and targets are unclear for the Bovine Skeletal Muscle Models.
-
Given that the training targets are unclear, using accuracy as the scorer/evaluation might be inefficient for hyperparameter tuning if the target values are imbalanced.
-
It is unclear whether there is information leakage between the train and test sets.
-
First paragraph in 2.2: it is unclear what the 15 models are and how they are different from each other.
-
It is unclear what “the original data model“ and “the initial model“ are.
-
Figure 2E: have the authors validate the expression levels of these 20 genes in different cell types by plotting them in the UMAP? The mean |SHAP| values are considerably small. It would be informative to show the SHAP values to see how each gene contributes to the prediction instead of only showing the mean |SHAP| values.
-
Section 2.3: why did the author choose the top 500 genes?
-
Figure 3C-F: I don’t see how these “UMAP results are similar to the parameter evaluations of the four models.” Can the authors offer a more numeric way to evaluate? For example, use a clustering method and evaluate how well different cell types are clustered together.
-
Figure 4B: SHAP values represent the contributions of each feature to the model prediction. It is unclear how summing SHAP values leads to the “importance ranking of 14 cell types.” Please elaborate on the biological interpretation of SHAP values.
-
Missing method descriptions:
-
The method section doesn’t include the information for “we ranked pathways based on gene signal intensity (Figure 3G)” and it is not obvious how this analysis was performed.
-
Figure 4E and other correlation-based results: how was the correlation calculated? What kind of correlation?
-
Figure 4FG and other enrichment analysis: The pathway enrichment analysis was not described in the method.
-
Section 2.6: please describe “model training”.
-
How do the authors define “metagenes”?
Minor comments:
-
Figure 4: SHAP values and Shapley values are different things. Do the authors really mean Shapley values?
-
Figure 6B: I don’t see any cow myofiber cells. This plot does not show expression levels, despite the text stating that “the expression of pig myofiber cells is significantly higher.”
Some sections lack smooth transitions. Enhancing the reasoning for the analysis will improve the readability and clarity.
Author Response
Thank you, reviewers, for reviewing this paper despite your busy schedules. After carefully considering your very professional questions and suggestions, the following is a response on behalf of all the co-authors of this study to your valuable comments and a description of the revised parts of the paper.
- The resolution of the figures is too low to read the text.
For some pictures with low resolution, we redrew them and replaced the original low-resolution pictures in the text. We set the resolution of the modified pictures to 600 dpi, hoping to meet the requirements of the reviewers and journal editors.
- The training scheme was not clearly described:
- The training features and targets are unclear for the Bovine Skeletal Muscle Models.
- Given that the training targets are unclear, using accuracy as the scorer/evaluation might be inefficient for hyperparameter tuning if the target values are imbalanced.
During the training process of the bovine skeletal muscle model, we set the features as genes and the target as 14 types of cells. Appropriately speaking, the model we trained is a classification model, that is, to classify cells based on gene expression. Before model training, we also speculated that due to the huge differences in the numbers among the 14 types of cells, the accuracy value of the model might not be high after evaluation. However, the training and testing results showed relatively high accuracy. We guessed whether over - fitting had occurred when we first used the original data to train the model, so subsequently we started to fit the data. But when using the fitted data and balancing the cell numbers, we found that the accuracy was even lower than that of the model trained with the unfitted data when we first started to fit the cells with a smaller number. As the fitting quantity increased, the accuracy of the model first approached that of the original data and then exceeded it. Therefore, we hypothesized that the model trained with the original data was under - fitted, and we selected the fitting parameters when the accuracy of the fitted data approached that of the original data. Subsequently, a series of our analyses verified the reasonableness of our hypothesis to a certain extent. The genes selected under this fitting parameter not only helped us screen out key cells and genes but also helped us redraw the single - cell map of bovine skeletal muscle.
- It is unclear whether there is information leakage between the train and test
We divided the training set and the test set in a ratio of 7:3. In the parameter section of the code, we conducted strict reviews and made regulations, so there is no problem of data leakage between the two datasets.
- First paragraph in 2.2: it is unclear what the 15 models are and how they are different from each other.
The parameter settings and hyperparameter settings for these 15 models during training are consistent. The only difference lies in the number of input cells. The first model uses the number of cells in the original data as input, which is more than 30,000. The second model is trained with data obtained by under - sampling the other 13 types of cells based on the number of neutrophils. That is, a model trained with 44*14 = 616 cells as input data. For the middle 12 models, they are trained after under - sampling the cells with a smaller quantity and over - sampling the cells with a larger quantity based on their own respective cell numbers. The 15th model is trained with data obtained by over - sampling other cells whose quantity is less than that of FAP. Specifically, it uses 11982*14 = 167748 cells as input data for training.
- It is unclear what “the original data model“ and “the initial model“ are.
The cause of this problem lies with us, and we have made the following modifications.
Original data model: The input data is a matrix of 19,090 genes * more than 32,700 cells. After training, the input of this model is the original unfitted data, and we call it the original data model.
Initial model: After fitting the data with the number of B cells, the accuracy, recall rate and other values of the trained model approach those of the original data model, but do not exceed them. Therefore, we choose it for subsequent analysis, so we call it the initial model.。
- Figure 2E: have the authors validate the expression levels of these 20 genes in different cell types by plotting them in the UMAP? The mean |SHAP| values are considerably small. It would be informative to show the SHAP values to see how each gene contributes to the prediction instead of only showing the mean |SHAP| values.
After much thought on this issue, we make the following statement.
Since our focus has not been on in - depth analysis in this aspect, we have not drawn the UMAP plots of these 20 genes in different cell types. First of all, when training the model, we input all the genes without removing some genes with low expression. This approach will, after SHAP interpretation, due to the introduction of non - important genes, lead to a slight marginalization of the SHAP values of important genes, and their contribution to a certain type of cell is uncertain. However, this approach also has a certain rationality. In both tissues and the organism, it is impossible for there to be genes that have no effect at all. Even genes with low expression levels may have some weak relationships with other important genes. If we blindly draw the UMAP plots of these 20 genes in 14 cell types, it is obviously somewhat unreasonable.
Regarding the problem that the |SHAP| values are quite small, it is actually the same as the answer in the previous paragraph. Taking the ACTA1 gene as an example, after the first SHAP interpretation, for the entire model, that is, the average |SHAP| value of the 14 types of cells combined, it is only close to 0.02, and for Myofiber cells, it is only about 0.006. However, after the subsequent second SHAP interpretation after selecting high - contribution genes, the SHAP value of the ACTA1 gene approaches 0.06, with an almost ten - fold increase in the change. It is still very small, but if we take the second SHAP value as the optimized result, we have improved the accuracy by ten times on the original basis. In my previous studies, I always thought that the larger the SHAP value, the better, and a very small SHAP value most likely indicated an error in the model. However, in subsequent studies, I found that this is not the case. What we need to focus on is not the magnitude of the SHAP value of a single feature. Because the more features a model has, the more likely it is to marginalize the SHAP value of each feature itself. At this step, we input more than 19,000 genes, that is, more than 19,000 features, which is equivalent to asking more than 19,000 people to divide a loaf of bread. Even if one person makes a high contribution, it is difficult for him to get half of the bread. And our subsequent SHAP interpretation is equivalent to dividing the loaf of bread among 476 people. Obviously, after the re - distribution, the share of these 476 people has increased.
Maybe my description is not very accurate and contains some of my own biases. I hope the reviewers can understand. Your question is very profound. The above is my answer after careful consideration based on my previous studies. Please forgive me if there are any offenses.
- Section 2.3: why did the author choose the top 500 genes?
In our application and learning of SHAP values, we read many articles in the relevant field. Regarding the selection of genes, there is no definite range for the number of genes. Some authors choose the top 20 genes, while others choose the top 100 genes. In response to this, we made two selections. First, after the first SHAP interpretation model, we selected the top 500 genes for 14 types of cells, that is, we selected 6000 genes. After analyzing these 6000 genes, we found that most genes appeared multiple times in our selection, and only more than 1000 genes had unique expressions. Then we filtered out the genes that appeared repeatedly in two or more types of cells and counted the genes that were co - expressed in the 14 types of cells and those uniquely expressed in each type of cell. The sum of them was 476 genes. Under such a selection, these genes have both the characteristics and the commonalities of the 14 types of cells.
The purpose of this process is to reconstruct the map of bovine skeletal muscle under the condition of a relatively small number of cells corresponding to a relatively small number of gene expressions. Subsequent analysis also verified that this purpose is feasible to a certain extent.
Actually, the above is mainly our exploration process. To a large extent, the purpose of selecting the top 500 genes is to avoid missing some secondary important information as much as possible. As answered in the previous question, we were worried that some genes with slightly higher rankings would be marginalized. Doing so greatly improves the rationality of our method.
- Figure 3C-F: I don’t see how these “UMAP results are similar to the parameter evaluations of the four models.” Can the authors offer a more numeric way to evaluate? For example, use a clustering method and evaluate how well different cell types are clustered together.。
Indeed, as the reviewer pointed out, since I'm not good at description myself, the description of these four result figures was too brief. We have made the following revisions.
It can be clearly observed from Figures 3C - F that, in terms of the degree of discrimination of each cell type after dimensionality reduction, for the vast majority of cells in Figures 3E and 3F, cells of the same type are clustered together. Although the clustering levels of several cell types in Figure 3F are not very obvious, it is evident that the 14 cell types are well - distinguished by the expression levels of the 476 - gene and 1161 - gene. In Figures 3C and 3D, in Figure 3C, four cell types in the middle area are too closely connected. In Figure 3D, only four cell types are clearly clustered, and it is difficult to distinguish the remaining ten cell types.
- Figure 4B: SHAP values represent the contributions of each feature to the model prediction. It is unclear how summing SHAP values leads to the “importance ranking of 14 cell types.” Please elaborate on the biological interpretation of SHAP values.
Thank you again for the reviewers' questions. Indeed, we didn't consider this issue at the beginning of our research. It was only after the completion of the paper writing, through discussions and reflections with my senior fellow apprentice that I found the answer to this question.
First of all, we need to understand the characteristics of SHAP values. One of them is additivity. After we quantify the expression matrix of the entire cell type and genes using SHAP values, each gene has SHAP values corresponding to 14 types of cells. For Figure 2E in the reviewers' previous question 3, it exactly shows the contribution of a certain gene to 14 types of cells. After quantifying the relationship between genes and different types of cells using SHAP values, we can regard the 14 types of cells as single cells of bovine skeletal muscle. In general, it is the ranking of the contribution of a single gene to skeletal muscle. Conversely, it is the contribution of a total of 476 genes to a certain type of cell. At this time, in general, it becomes the ranking of the contribution of a single cell type to skeletal muscle.
- Missing method descriptions:
- The method section doesn’t include the information for “we ranked pathways based on gene signal intensity (Figure 3G)” and it is not obvious how this analysis was performed.
Here, we really didn't describe it clearly in the material.
This figure was obtained by us after visualizing it on the online analysis website STRING. After entering the gene names and selecting cattle as the reference species, in the parameters of the step before visualization, we chose to visualize it based on the signal intensity.
- Figure 4E and other correlation-based results: how was the correlation calculated? What kind of correlation?
Calculate the Pearson correlation coefficients between genes using the `corr()` method of pandas to obtain a correlation matrix named `gene_correlation_matrix`. Each element in the matrix represents the linear correlation between the corresponding two genes.
- Figure 4FG and other enrichment analysis: The pathway enrichment analysis was not described in the method.。
In the revised article, we made the following additions:
This analysis still uses the STRING online website. The inputs are still the species name and the selected gene name.
- Section 2.6: please describe “model training”.
For the model training in this process, we used the default parameters on its GitHub. During the process of reproducing this model, we also encountered some minor issues. By searching the questions on the forum where those who reproduced this model asked the original author, we also found that some readers asked whether the essential parameters of the model training script needed to be modified. The original author gave an affirmative reply, saying that there was no need to forcefully modify them. These parameters have been trained on multiple datasets and are somewhat reliable. We only need to modify the number of selected hypervariable genes and the number of metagenes when finally constructing the metagene space.
- How do the authors define “metagenes”?
The concept of metagene is not defined by us. It was proposed by the authors of the SATURN protein language model. I will describe the full original text in the following paragraph as follows:
The major challenge of cross-species integration is that different datasets have different genes that may not have common one-to-one homologs. Subsetting each species’ set of genes to the common set of one-to-one homologs leads to losing a large portion of biologically relevant genes. Increasing the number of species exacerbates this problem, as a gene must have a homolog in each species to be considered for integration. SATURN overcomes this problem by using large protein language models to learn cell embeddings that encode the biological meaning of genes. SATURN maps cross-species datasets in the space of functionally related genes determined by protein embeddings. SATURN’s use of protein language models allows it to represent functional similarities even between remotely homologous genes that are missed by integration methods that rely on sequence-based similarity.
In particular, SATURN integrates scRNA-seq datasets generated from different species with different genes by mapping them to a joint low-dimensional embedding space using gene expression and protein representations. SATURN takes as input: (i) scRNA-seq count data from one or multiple species, (ii) protein embeddings generated by a large protein embedding language model like ESM2, and (iii) initial within-species cell annotations (from cell-type assignments if available or obtained by running a clustering algorithm). The language model takes a sequence of amino acids and produces a protein representation vector. Given gene expression and protein embeddings, SATURN learns an interpretable feature space shared between multiple species. We refer to this space as a macrogene space and it represents a joint space composed of genes inferred to be functionally related based on the similarity of their protein embeddings. The importance of a gene to a macrogene is defined by a neural network weight—the stronger the importance, the higher the value of the weight that connects the gene to the macrogene.
Given the shared macrogene expression space across different species, SATURN then learns to represent cells across multiple species as nonlinear combinations of macrogenes. The neural network in SATURN is first pretrained with an autoencoder with zero inflated negative binomial loss, regularized to reconstruct protein embedding similarities using gene-to-macrogene weights (Methods). Using the pretrained network as initialization, SATURN then learns a mapping of all cells to the shared embedding space with a weakly supervised metric learning objective. This allows SATURN to calibrate distances in the embedding space to reflect cell label similarity. In particular, the objective function in SATURN consists of two main components: (i) forcing different cells within the same dataset far apart using weak supervision; and (ii) forcing similar cells across datasets close to each other in an unsupervised manner (Methods). This objective enables SATURN to integrate cells across different species, while preserving cell-type information within each species’ dataset.
- Figure 4: SHAP values and Shapley values are different things. Do the authors really mean Shapley values?
This is due to our lack of proficiency in professional knowledge. Our original intention was the SHAP value. Thank the reviewers for pointing out the problem.。
- Figure 6B: I don’t see any cow myofiber cells. This plot does not show expression levels, despite the text stating that “the expression of pig myofiber cells is significantly higher.”
Due to our oversight, we failed to present the porcine skeletal muscle data. We suspect that the excessive number of Myofiber cells in the porcine skeletal muscle data might have led to this result. For the porcine skeletal muscle data, the number of Myofiber cells is 30,328, while there are only more than 500 in bovines. We guess that the excessive number of these cells in pigs might have masked the expression of these cells in bovines. In our analysis, we ran this model three times, and the results are still as shown in Figure 6B. The following is the porcine skeletal muscle data. We have added the details of the porcine skeletal muscle data cells in the supplementary table.
In the subsequent submitted revisions, we further discussed this issue.
Reviewer 2 Report
Comments and Suggestions for Authors
This manuscript presents an innovative integration of machine learning, SHAP (Shapley Additive Explanations) analysis, and cross‐species transcriptomic comparisons to identify key cell types and genes in bovine skeletal muscle. The study’s approach—especially the reconstruction of a gene importance matrix via SHAP values and the use of the SATURN protein language model for cross-species alignment—is novel.
Materials and Methods:
Provide detailed hyperparameter ranges and settings: further details regarding the hyperparameter search (e.g., parameter ranges, the number of iterations, and stopping criteria) would enhance the reproducibility. how the sampling strategies were optimized and how potential overfitting was monitored during model training?
Clearly describe the steps involved in data normalization, quality control, and any feature selection processes prior to model training. Including details on how the raw single-cell data were processed will help others replicate your workflow. Specify the operating system, hardware specifications, and software dependencies (with version numbers) used throughout the study.
The decision to focus on the 476 genes reconstructed from SHAP values is central to the study. Please elaborate on the rationale behind choosing this specific threshold and provide further statistical justification for this cut-off.
A more detailed description of the alignment process, including any challenges encountered and how differences in gene expression profiles were statistically validated in the use of the SATURN protein language model for aligning.
While the internal validation via cross-validation is thorough, the robustness of the findings could be enhanced by testing the model on an independent external dataset.
Results:
Several figures need a higher resolution and clearer labeling. For example:
Figure 1: The abbreviations in the figure, such as FAP, have no explanations in the figure caption.
Figure 2: The resolution is too low. The words in the figure are completely illegible.
Figure 3: The resolution is too low. The words in the figure are completely illegible.
Clearly specify which statistical tests were used for each analysis. Provide explicit p-values, confidence intervals (CIs), and effect sizes for key comparisons and correlations.
Discussion:
As noted briefly in the discussion, single-cell sequencing technologies, such as 10x sequencing, often suffer from data loss, capturing only a fraction of the cellular information. This limitation should be emphasized in the discussion, with further clarification on how much data might be lost in the current study and its potential impact on the conclusions.
The limitations of the study, such as potential biases from sampling techniques, overfitting in machine learning models, and the constraints of single-cell sequencing, should be discussed in more detail.
Author Response
First of all, I would like to thank the reviewers for reviewing this paper. On behalf of all the authors of this paper, I will provide a unified response to the reviewers' comments.
- Provide detailed hyperparameter ranges and settings: further details regarding the hyperparameter search (e.g., parameter ranges, the number of iterations, and stopping criteria) would enhance the reproducibility. how the sampling strategies were optimized and how potential overfitting was monitored during model training?
The following are the entries for the model parameters and hyperparameter settings: First, to ensure the reproducibility of the experimental results, during the process of dividing the training set and the test set and the training process of the model, we selected the random seed as 42. When using RandomizedSearchCV to tune the hyperparameters of the random forest model rf_model, the following hyperparameter distributions were defined. n_estimators randomly selects an integer from 100 to 500, which represents the number of decision trees in the random forest; max_features can take the values of 'auto','sqrt' or 'log2', used to determine the calculation method of the maximum number of features considered when constructing each tree; max_depth randomly selects an integer from 10 to 50, which controls the maximum depth of the decision tree; min_samples_split randomly selects an integer from 2 to 10, representing the minimum number of samples required to split an internal node; min_samples_leaf randomly selects an integer from 1 to 4, that is, the minimum number of samples required for a leaf node; bootstrap is a boolean value that can take the values of True or False, used to decide whether to use bootstrapping when constructing the tree. Based on these hyperparameter distributions, a RandomizedSearchCV object was created, the number of iterations of the random search was set to 20 times, and 5 - fold cross - validation was adopted.
To monitor the potential overfitting of the model, we designed a set of processes. First, we trained a random forest model on the downloaded standardized raw data. Then we regarded values such as its accuracy and precision as the standard. Since the dataset we used contains 14 types of cells and their numbers are different, we adopted the following sampling strategies for these 14 types of cells. If the cells are arranged in ascending order of the number of each type, for the cell ranked first, the 13 types of cells with a larger number than it are all undersampled in order to make their numbers tend to be the same as it, so as to eliminate biological differences. For the cell ranked 14th, the cells with a smaller number than it are all oversampled. For the cells in the middle rankings, taking the cell ranked 7th as an example, the 6 types of cells with a smaller number than it are oversampled, and the 7 types of cells with a larger number than it are undersampled. Of course, such an approach obviously introduces too much synthetic data. For this reason, we also evaluated the accuracy and recall rate of the models trained in the last 14 times and compared them with the model trained on the raw data as the standard. Since the synthetic data depends on the raw data, if the amount of synthetic data is too large, the model will naturally overfit. Therefore, we selected a fitting model that is close to the accuracy and recall rate evaluation of the model trained on the raw data, so that the accuracy of the model trained on the selected fitting data is not higher than that of the model trained on the raw data. In this way, on the premise of using sampling, we solved the problem of model overfitting. Subsequently, we also discussed the functions of the key genes selected after model interpretation to prove the rationality of our idea.
- Clearly describe the steps involved in data normalization, quality control, and any feature selection processes prior to model training. Including details on how the raw single-cell data were processed will help others replicate your workflow. Specify the operating system, hardware specifications, and software dependencies (with version numbers) used throughout the study.
During the process of training the random forest model for bovine skeletal muscle, the data we used was sourced from the public data in the GEO database. The authors of the data uploaded the processed single - cell matrix. Therefore, we did not perform any processing on the bovine skeletal muscle data. In the materials section, we marked the source of the data. I believe that the authors who provided the data described the processing process more professionally than I could.
Of course, during the model training process, we input more than 19,000 genes in total. Obviously, this introduced some unnecessary genes, which might cause potential differences. In response, our study specifically used SHAP to interpret the model twice. First, we quantified the contribution of all genes to each cell type. Then, we set a threshold of the top 500 genes ranked by SHAP values for each cell type to screen out some potentially important genes. After that, we used SHAP to interpret them again, so as to further quantify the contribution of relatively important genes to cells.
Regarding the operating system, hardware specifications, and software dependencies in the entire study, our description in the materials section might not be appropriate, so we made some modifications. For the construction of the bovine skeletal muscle model, the training of the first 14 skeletal muscle models was carried out on a laptop with a Windows 11 system, 64GB of running memory, a 12th - generation i7 processor, and an NVIDIA 4060 standard graphics card. Due to the excessive number of cell fittings, the 15th model was trained on a server with 128GB of running memory. This is the hardware level we used. Of course, this is not a definite condition. If a lower - level hardware is selected, the training might take too long, but corresponding results can still be obtained. The SATURN protein model was also trained on the above - mentioned laptop. We have described in detail the software required for bovine skeletal muscle training in the materials section. It was implemented in the Python language, and we have indicated all the required Python packages and their versions. As for the Python packages and versions required by SATURN, interested readers can read its official website by themselves, as the authors have fully described these issues. The only thing we would like to add is that if readers have an NVIDIA graphics card, they can download CUDA and CUDNN to use the local GPU to accelerate the training and calculation process. If not, they need to delete the number of GPUs and the part of the code that uses the GPU in the script downloaded from its official website and the code for the model training process.
- The decision to focus on the 476 genes reconstructed from SHAP values is central to the study. Please elaborate on the rationale behind choosing this specific threshold and provide further statistical justification for this cut-off.
Selecting 476 genes is not a statistical criterion but rather a result of our study. During our application and learning of SHAP values, we read a large number of articles in the relevant field. Regarding the selection of genes, there is no definite range for the number of genes. Some authors choose the top 20 genes, while others choose the top 100 genes. In response to this, we made two rounds of selections. First, after the first SHAP interpretation model, we selected the top 500 genes from 14 types of cells, which means we selected 6000 genes in total. After analyzing these 6000 genes, we found that most genes appeared multiple times in our selections, and only more than 1000 genes had unique expression. Then, we filtered out the genes that appeared repeatedly in two or more types of cells and counted the genes that were co - expressed among the 14 types of cells and those that were uniquely expressed in each type of cell. The sum of these genes was 476. Under such a selection, these genes possess both the characteristics specific to each of the 14 types of cells and the commonalities shared among them.
The purpose of this process was to reconstruct the bovine skeletal muscle map under the condition of a relatively small number of cells corresponding to a relatively small number of gene expressions. Subsequent analyses also verified the feasibility of this purpose to a certain extent.
- A more detailed description of the alignment process, including any challenges encountered and how differences in gene expression profiles were statistically validated in the use of the SATURN protein language model for aligning.
Thanks to the reviewers for their questions about this process. We have made revisions in the original text, and the following is the unified response.
We followed the species protein sequence training script on its official website and in-put the protein sequences of cattle and pigs to train the corresponding protein models for the two species. Then, we used the official training script of SATURN to train the model with the generated protein embeddings from the training, the gene expression matrices of the two species, and the initial intra - species cell annotations. Did not change the parameters of the original script in both training sessions. We set the num-ber of highly variable genes to 8000 and the number of macrogenes to 2000.
- While the internal validation via cross-validation is thorough, the robustness of the findings could be enhanced by testing the model on an independent external dataset.
After discussion, we very much agree with this opinion of the reviewers. In future research, we will try this approach and then analyze the results obtained and the new problems that arise.
- Several figures need a higher resolution and clearer labeling. For example:
Figure 1: The abbreviations in the figure, such as FAP, have no explanations in the figure caption.
Figure 2: The resolution is too low. The words in the figure are completely illegible.
Figure 3: The resolution is too low. The words in the figure are completely illegible.
For the cell names that were not abbreviated correctly, we have made corrections in the main text. We have modified and re - uploaded the low - resolution pictures.
- Clearly specify which statistical tests were used for each analysis. Provide explicit p-values, confidence intervals (CIs), and effect sizes for key comparisons and correlations.
In this study, we did not use many statistical tests, except for the statistical test parameters such as the F1 score, precision, and recall rate in the first draft.Before visualizing the correlation relationships between the selected genes, we calculated the Pearson correlation coefficients between the genes using the corr() method in pandas and obtained a correlation matrix named gene_correlation_matrix. Each element in the matrix represents the linear correlation between the correspond-ing two genes In this study, a gene - to - gene correlation greater than 0.8 is defined as a high correlation
- As noted briefly in the discussion, single-cell sequencing technologies, such as 10x sequencing, often suffer from data loss, capturing only a fraction of the cellular information. This limitation should be emphasized in the discussion, with further clarification on how much data might be lost in the current study and its potential impact on the conclusions.
The limitations of the study, such as potential biases from sampling techniques, overfitting in machine learning models, and the constraints of single-cell sequencing, should be discussed in more detail.
We would like to express our gratitude to the reviewers for their valuable suggestions in the discussion section. We will provide a unified response to the following two suggestions here, and the relevant parts have been revised in the discussion section as described below.
In this study, we comprehensively explored bovine skeletal muscle single - cell transcriptomic data through the integration of explainable machine learning models with open - source protein large models, and successfully conducted a cross - species comparison with porcine skeletal muscle single - cell transcriptomes. It is crucial to emphasize that the current mainstream single - cell sequencing technologies, repre-sented by 10× single - cell sequencing, have significant limitations in information ex-traction. These technologies are only capable of capturing approximately 30% - 40% of the cellular information, which means that at least 60% of the essential data is lost [26, 27]. This substantial data loss not only severely degrades the data quality but also makes the information obtained from less - expressed cell types even sparser [28]. In the context of our current research, such a high proportion of data loss implies that a large amount of potentially valuable biological information may be overlooked. This could have a profound impact on our conclusions, as the missing data might contain key regulatory factors, rare cell populations, or subtle expression patterns that are es-sential for a comprehensive understanding of the biological processes in bovine and porcine skeletal muscles. Therefore, while our findings are significant, they should be interpreted with caution, taking into account the inherent limitations of the single - cell sequencing technology employed.
Although this study has achieved certain results, there are limitations that cannot be ignored. In terms of sampling technology, there may be potential biases. The sample selection process may not have fully covered the diversity of the target population, resulting in insufficient representativeness. This may limit the generalizability of the research results and prevent them from accurately reflecting the overall situation.The machine - learning model has the risk of overfitting. The model performs well on the training set and test set data, but it is very likely to have poor predictive ability for new data, which may weaken the reliability and practicality of the research conclusions.The single - cell sequencing technology itself also has its limitations. For example, differences in cell capture efficiency may lead to some cell types not being fully detected. Uneven sequencing depth may affect the accuracy of gene expression quantification. Moreover, the high cost of this technology and the complexity of data analysis also restrict the breadth and depth of the research to a certain extent.Subsequent research needs to focus on improving these issues to enhance the quality of the research and the credibility of the results.